# The Effect of Selected Bee Products on Adhesion and Biofilm of *Clostridioides difficile* Strains Belonging to Different Ribotypes

**DOI:** 10.3390/molecules27217385

**Published:** 2022-10-30

**Authors:** Dorota Wultańska, Bohdan Paterczyk, Julita Nowakowska, Hanna Pituch

**Affiliations:** 1Department of Medical Microbiology, Medical University of Warsaw, 02-004 Warsaw, Poland; 2Imaging Laboratory, Faculty of Biology, University of Warsaw, 02-096 Warsaw, Poland

**Keywords:** *Clostridioides difficile*, PCR-ribotypes, adhesion, biofilm, bee products

## Abstract

There is an ongoing search for alternative treatments for *Clostridioides difficile* infections. The aim of the study was to investigate the antibacterial and antibiotic activity of bee products against *C. difficile* strains with different polymerase chain reaction ribotypes (RTs). The minimum inhibitory concentration (MICs) of Manuka honey 550+, goldenrod honey, pine honey, and bee bread were determined by the broth dilution method. *C. difficile* adhesion to HT-29, HT-29 MTX, and CCD 841 CoN cell lines was assessed. Biofilm was cultured in titration plates and visualized by confocal microscopy. The MICs of Manuka honey for *C. difficile* 630 and ATCC 9689 strains and control strain, M 120, were 6.25%, 6.25%, and 1.56% (*v*/*v*), respectively; of goldenrod honey, 50%, 50%, and 12.5%, respectively; of pine honey, 25%, 25%, and 25%, respectively; and of bee bread, 100 mg/L, 50 mg/L, and 100 mg/L, respectively. Manuka honey (1%) increased adhesion of *C. difficile* RT176 strains, and one strain of RT023, to the CCD 841 cell line. Pine honey (1%) increased RT027 adhesion to the HT-29 cell line. Manuka honey, pine honey, and bee bread at subinhibitory concentrations increased the adhesion of *C. difficile*. Our research proved that bee products are active against the tested strains of *C. difficile.*

## 1. Introduction

*Clostridioides difficile* (previously *Clostridium difficile*; *C. difficile*) is an anaerobic gram-positive rod that produces highly resistant spores capable of spreading in the environment. Over the past four decades, *C. difficile* has become the leading cause of antibiotic-associated diarrhea worldwide, imposing a heavy burden on the healthcare system. The histopathological hallmark of *C. difficile* infection (CDI) is damage to the colonic epithelial cells with the accompanying acute inflammatory response and the formation of pseudo-membranes. *C. difficile* is a multi-drug resistant microorganism. Until recently, standard therapy for CDI included vancomycin and metronidazole. However, while active against vegetative organisms, these antibiotics do not effectively inhibit spore formation. Currently, fidaxomicin and vancomycin are preferred as standard of care [1]. Antibiotics such as ampicillin, amoxicillin, cephalosporins, clindamycin, and fluoroquinolones disrupt the healthy microflora in the digestive tract, which leads to proliferation of *C. difficile* [2].

Bacterial adhesion to the host’s biological membranes is the first step in colonization and biofilm formation. It determines the long-term presence of pathogens in the body and their impact on human health. The ability of *C. difficile* to form biofilm has significantly contributed to the increasing antibiotic resistance [3,4]. In fact, according to Semenyuk et. al. [5], *C. difficile* biofilms cause a 100-fold increase in resistance to metronidazole [5]. Although little information has been published so far on the ability of *C. difficile* to form biofilm in vivo, it is well known that multicellular structures can potentially protect bacteria against cellular immune response and antibiotic activity [3,4]. Moreover, CDI relapse can be associated with the persistence of biofilm [4].

In the last decade, several hypervirulent strains of *C. difficile* have emerged. The most prominent strain has been classified as polymerase chain reaction (PCR) RT027, which appeared mainly in Canada, North America, and several European countries [6,7,8,9,10,11]. The epidemics of *C. difficile* RT027 infections have been also reported in Poland [12,13]. The investigation of epidemics from those countries showed that RT027 is associated with a worldwide increase in hospital outbreaks, characterized by the recurrence of infections and a high mortality rate [7,14,15,16]. In addition, recent research showed that RT027 strains still have a significant contribution to the occurrence of CDIs [17,18,19] and are associated with a heavy clinical, therapeutic, and epidemiological burden. The first Polish isolate of *C. difficile* RT027 was detected in 2005, and the closely related RT176 was discovered in 2008 [12,20]. CDI outbreaks associated with RT027 and RT176 were documented in three hospitals in Poland in the years 2008–2010 [21]. *C. difficile* RT023 is associated with severe infections similar to those seen with “hypervirulent” strains from clade 2 (RT027) and 5 (RT078), with particularly bloody diarrhea [22]. Considering that currently used drugs for CDI are not always effective, there is a need to review novel therapeutic agents, including substances of natural origin, that could supplement the standard treatment regimen. 

The treatment of infections associated with antibiotic resistance is challenging due to adverse effects, limited treatment options, and the increasing tendency to antibiotic resistance. Healthcare facilities are at particular risk of infections. Therefore, there is an urgent need for alternative therapeutic interventions that would be both effective and free from adverse effects. One such intervention is the use of bee products, such as honey, herb-infused honey, pollen, bee bread, propolis, and others [23]. 

Honey has been used as a therapeutic agent for centuries. However, it is only recently that the antibacterial activity of honey has been demonstrated in vitro and in vivo [24,25,26,27,28,29,30,31]. Natural honey is derived from the nectar collected by honeybees. High sugar content, combined with low pH, bee enzymes, bee peptides, and phytochemicals, contributes to the antibacterial activity of honey [32,33,34,35,36,37,38]. Moreover, honey has antioxidant, anti-inflammatory, and antihyaluronidase properties, which vary depending on the nectar source [32,39,40]. The amount of natural phenols in honey determines its inhibitory effect [39,40]. Honey with high phenol levels has a greater inhibitory effect than honey with no or low phenol content.

Honey is a natural food product that, apart from its nutritional value, has valuable healing properties thanks to the presence of bioactive ingredients. In general, biologically active compounds contained in honey can be divided into two groups: antibacterial and antioxidant [41,42]. However, these two factors influence each other, and their combination enhances the health-promoting properties of honey. Honey has bacteriostatic and bactericidal activity against several human pathogens, especially gram-positive bacteria such as *Staphylococcus aureus* and gram-negative such as *Escherichia coli* and *Pseudomonas* spp. [43].

The composition of honey depends primarily on the country of origin, but seasonal and environmental factors are also of great importance. As a result, the chemical composition of honey varies significantly, and different types of honey show distinct health-promoting properties [41,42].

Among the phenolic compounds of honey, it is mainly antioxidants that have been widely studied. More than 150 phenolic compounds in honey have been tested, including phenolic acids and flavonoids [44]. These compounds are classified as the thermostable components and are not sensitive to elevated temperatures [45]. Moreover, it should be noted that, in addition to antioxidant activity, they show bactericidal, anti-inflammatory, antiallergic, anticoagulant, and antitumor properties [46]. The aim of this study was to investigate the antimicrobial and antibiofilm activity of selected bee products against *C. difficile* strains belonging to different PCR-RTs.

## 2. Results

### 2.1. MIC and MBC Values of C. difficile with Bee Products

The results are summarized in Table 1. The MIC of Manuka honey for *C. difficile* strains 630, ATCC 9689, and M 120 was 6.25%, 6.25%, and 1.56% (*v*/*v*), respectively, and the MBC was 6.25%, 6.25%, and 3.125% (*v*/*v*), respectively. The MIC of goldenrod honey was 50%, 50%, and 12.5%, respectively; of pine honey, 25%, 25%, and 25%, respectively; and of bee bread, 100 mg/L, 50 mg/L, and 100 mg/L, respectively.

The reference strains did not show MBC resistance values under the influence of goldenrod honey, pine honey, or bee bread. The MIC value of clinical strains with given RTs was different for the bee products tested and ranged from 1.56% to 6.25% for Manuka honey, from 1.56% to 50% for goldenrod honey, from 1.56% to 25% for pine honey, and from 25 mg/L to 100 mg/L for bee bread. The MBC values of clinical *C. difficile* strains for Manuka honey ranged from 12.5% to 3.125%. The MBC values for RT176 strains and one RT027 strain under the influence of goldenrod honey ranged from 50% to 25%, and under the influence of pine honey, from 50% to 12.5%. On the other hand, MBC values for RT023 strains and one RT027 strain were above 100% under the influence of goldenrod honey and pine honey. All clinical strains were resistant to bee bread, and the MBC value was above 200 mg/L (Table 1).

### 2.2. Effect of Bee Products on Adhesive Properties of C. difficile

The tested reference strains 630 and M 120 and clinical strains 1468 (RT027), 25694 (RT023), and 2628 (RT176) showed adhesion to the CCD 841, HT-29, and HT-29 MTX cell lines, and with addition of 1% of tested bee products. Manuka honey increased the adhesion of strain RT176 and strain RT023 to the CCD 841 cell line and that of strain RT023 to the HT-29 MTX cell line (*p* < 0.05). Pine honey increased the adhesion of strain RT027 to the HT-29 cell line (*p* < 0.05). Finally, bee bread increased the adhesion of strain M 120 to the HT-29 cell line (*p* < 0.05) (Figure 1).

### 2.3. Effect of Bee Products on C. difficile Biofilm Formation

Manuka honey, at all tested concentrations of 100%, 50%, 25%, 12.5%, 6.25%, 3.125%, and 1.56%, had a significant effect on the formation of biofilm biomass by reference strains 630, ATCC 9689, M 120, and one clinical strain 536/12 (RT176). For the clinical strains 1468/12 (RT027), 2628/12 (RT176), and 1974/12 (RT176), Manuka honey did not affect the formation of biofilm biomass (at any concentration). For the remaining clinical strains, including the two tested with RT023, biofilm biomass formation was significantly affected by higher concentrations of honey ranging from 100% to 6.25% (Table 2).

Under the influence of goldenrod honey, biofilm biomass formation was significantly higher in the concentration range of 100% to 12.5% for the reference strains and clinical strains 1468/12 (RT027), 5291/12, and 2628/12 (RT176). For strain 536/12 (RT176), biofilm biomass formation turned out to be significant at low concentrations (1.56%–12.5%) and a single high concentration of 100%. The biofilm biomass formation was not significantly affected for strains with RT023 (except for the concentration of 6.25% in one strain) and for one strain 1974/12 (RT176) (Table 3).

Pine honey in the concentrations of 100% to 12.5% had a significant effect on the formation of biofilm biomass by the reference strains. Pine honey at all concentrations (1.56%–100%) significantly influenced the formation of biofilm biomass by 1468/12 (RT027) strain. The tested product had a significant effect on biofilm biomass formation in the concentration range of 25% to 3.125% by the 536/12 (RT176) strain. In two clinical strains, 5291/12 and 3136/12 (RT176), pine honey did not have a significant effect on the formation of biofilm biomass. In the remaining clinical strains, pine honey significantly increased the formation of biofilm biomass only at single concentrations: for strain 2292 (RT027) at concentrations of 3.125% and 1.56%; for strains 2333/06 and 25694/12 (RT023) at concentrations of 100% and 6.25%, respectively; and for strains 2628/12 and 1974/12 (RT176) at concentrations of 3.125%, 1.56%, and 12.5% (Table 4).

Bee bread had a significant effect on the biofilm formation of reference strain 630 at concentrations of 200 mg/L and 100 mg/L; ATCC 9689 at concentrations of 200 mg/L, 100 mg/L, and 50 mg/L; and strain M 120 at concentrations of 200 mg/L, 50 mg/L, and 12.5 mg/L. Bee bread had no significant effect on the formation of biofilm biomass by clinical strains 1468/12 and 2292 (RT027), strain 25694/12 (RT023), and strains 536/12 and 1974/12 (RT176). For the remaining clinical strains, bee bread significantly affected the formation of biofilm biomass at single concentrations: for clinical strain 2333/06 with RT023 at 200 mg/L, and for strains 5291/12, 2628/12, and 3136/12 (with RT176) at 100 mg/L, 100 mg/L, and 50 mg/L, respectively (Table 5).

### 2.4. Confocal Laser Scanning Microscopy

Reference strain 630 produced an irregular, heterogeneous biofilm with sparse and thin three-dimensional (3D) architecture. Under the subinhibitory concentration of Manuka honey 550+ (3.125%), the biofilm was regular and sparse, with a high 3D architecture containing microaggregates and significantly elongated plankton cells. Under the influence of a subinhibitory dose of goldenrod honey (25%), the biofilm was irregular and sparse, with high 3D architecture containing microaggregates and short plankton cells. Under the influence of a subinhibitory dose of pine honey (12.5%), the biofilm was irregular and heterogeneous, with a thick 3D architecture with very short, twisted plankton cells and tightly packed microaggregates in places. Under the influence of a subinhibitory concentration of bee bread (50%), the biofilm was regular and sparse, with a medium-high 3D architecture containing a small number of microaggregates and very elongated plankton cells.

The reference strain M 120 produced a homogeneous, regular biofilm with a dense architecture with microaggregates, composed of cells tightly arranged next to each other along the entire width. Under the subinhibitory concentration of Manuka 550+ honey (0.78%), the biofilm was homogeneous, dense, carpet-like, and composed of tightly packed and contiguously arranged microaggregates. Under the influence of a subinhibitory concentration of goldenrod honey (6.25%), the biofilm was very thick, regular, carpet-like, and consisted of tightly packed and contiguously arranged microaggregates. Under the subinhibitory concentration of pine herb honey (12.5%), the biofilm was very thick, regular, carpet-like, and consisted of tightly packed and contiguously arranged microaggregates with high 3D architecture, irregularly shaped, with significantly elongated plankton cells and microaggregates.

The biofilm of the clinical strain with RT027 (No 4. 1468/12) showed irregular, heterogeneous, thin, and sparse architecture of both microaggregates and cells. Under the influence of a subinhibitory concentration of Manuka 550+ honey (3.125%), the biofilm was thin, irregular, heterogeneous, and dense with the architecture of microaggregates. Under the influence of a subinhibitory concentration of goldenrod honey (6.25%), the biofilm was homogeneous, dense, carpet-like, but thin with tightly packed microaggregates. Under the influence of a subinhibitory dose of pine herb honey (1.56%), the biofilm was regular, dense, with a low 3D architecture containing tightly packed microaggregates. Under the influence of a subinhibitory concentration of bee bread (25%), the biofilm was homogeneous and thin, with large amounts of plankton cells.

The biofilm of the clinical strain RT023 (No 7. 25694/12) was irregular, heterogeneous, and sparse in both microaggregate and cell architecture. Under the influence of a subinhibitory dose of Manuka 550+ honey (3.125%), the biofilm was regular, with low 3D architecture containing microaggregates. Under a subinhibitory dose of goldenrod honey (25%), the biofilm was homogeneous, dense, and carpet-like, with tightly packed microaggregates and cells, but with a low 3D architecture. Under the influence of a subinhibitory dose of bee bread (12.5%), the biofilm was regular with microaggregates and very short plankton cells, with a thin 3D architecture.

The biofilm of the clinical strain RT176 (No 9. 2628/12) was irregular, heterogeneous, and sparse in the architecture of both microaggregates and cells. Under the influence of a subinhibitory dose of Manuka 550+ honey (1.5625%), the biofilm had a very thin 3D architecture containing few microaggregates and cells. Under the influence of a subinhibitory dose of goldenrod honey (1.5625%), the biofilm was homogeneous, dense, and carpet-like with lots of regular plankton cells, with low 3D architecture. Under the influence of a subinhibitory dose of pine herb honey (0.78%), the biofilm was very sparse, irregular, with a small number of microaggregates and plankton cells, with a very thin 3D architecture. Under the influence of a subinhibitory dose of bee bread (50%), the biofilm was regular, sparse, with many short plankton cells and with low 3D architecture (Figure 2).

## 3. Discussion

Honey, which also stimulates inflammatory cytokines [47], has been identified as a potent superoxide anion scavenger and a highly effective reactive oxygen species inhibitor that is released from human neutrophils [38]. Evidence from published studies shows that honey destroys *Pseudomonas aeruginosa* cell walls [48] and disrupts cell division in methicillin-resistant *S. aureus* [49]. Honey has been shown to have a broad spectrum of antibacterial activity against many pathogens, including antibiotic-resistant organisms and their biofilms [50,51].

Both raw and sterilized (pasteurized or irradiated) honey showed the desired broad spectrum of activity against gram-positive and gram-negative bacteria, including clinically important pathogens such as *Burkholderia cepacia* [52], *P. aeruginosa* [31,53], *Salmonella enterica*, *Serratia marcescens*, *E. coli*, *Klebsiella pneumoniae* [54], *S. aureus*, *Salmonella typhimurium*, *Shigella sonnei* [55], and *Streptococcus pyogenes* [56]. In addition, several multidrug-resistant bacteria, including methicillin-resistant *S. aureus* and vancomycin-resistant Enterococci, were susceptible to honey [57]. Research on activity-related biomarkers has shown that hydrogen peroxide and methylglyoxal contribute significantly to the antibacterial activity of honey [58]. However, neither the action of hydrogen peroxide nor methylglyoxal can account for the total antibacterial activity of the honey since their removal did not completely abolish its cytotoxicity. It is an undeniable fact that almost all types of honey exhibit at least bacteriostatic activity, regardless of their botanical or geographical origin, which makes the antibacterial activity an inherent property of honey and a desirable source for drug development [59].

In the presented study, we did not observe antiadhesive action by the selected bee products. Instead, we observed an increase in adhesion under 1% concentration of bee products. It is hypothesized that this may be due to the low concentrations of tested substances. According to Mizzi et al., high sugar concentrations inhibit bacterial growth, while low concentrations stimulate bacterial growth. The same may apply to bacterial adhesion [60].

In our previous study, RT027 strains showed the highest potential for biofilm formation compared with other PCR-RTs [28]. *C. difficile* strains belonging to RT017, RT023, RT027, and RT046 were completely inhibited by Manuka honey MGO 400+ at a concentration of 6.25%. Both the MIC and MBC values were found to be 6.25%. The study demonstrated that Manuka honey MGO 400+ could inhibit biofilm formation in vitro in clinically significant strains of *C. difficile*, especially those belonging to RT027 [28]. Manuka honey, derived from the Manuka tree (*Leptospermum scoparium*), is known to exhibit antimicrobial properties that are associated with its significant content of methylglyoxal, a natural antibiotic. In our study, the MIC and MBC values for Manuka honey 550+ were 6.25% and were comparable with the values observed for strains with RT027 and RT023 in the previous study. For RT176, these values ranged from 1.56% to 12.5%. Under the influence of high concentrations (6.25–100%), Manuka Honey 550+ inhibited the formation of biofilm in strains belonging to RT027, RT023, and some RT176. Goldenrod honey, under the influence of 50% and 100% concentrations, inhibited the biofilm formation of clinical strains. Pine honeydew biofilm formation was significantly inhibited at all concentrations in the tested clinical strains, except for one strain RT176, RT023 and RT027. Pine herb honey also changed the appearance of plankton cells. Bee bread did not enhance or inhibit biofilm formation. Our findings confirm the previous literature [61,62,63].

The limitation of the presented study is a lack of information on the phytochemical composition of tested products.

## 4. Materials and Methods

### 4.1. Bacterial Strains

For this study, the following reference strains of *C. difficile* were used: 630 (RT012), ATCC 9689 (RT001), and M 120 (RT078). Moreover, nine clinical strains of C. difficile were used: 1468/12 and 2292 (RT027); 2333/06 and 25694/12 (RT023); 5291/12, 2628/12, 536/12, 1974/12, and 3136/12 (RT176).

The clinical strains of *C. difficile* were isolated from Polish hospitals. All *C. difficile* strains were obtained from the collection of the Anaerobic Laboratory of the Chair and Department of Medical Microbiology, Medical University of Warsaw, Poland. The strains were stored in the Microbank™ bacterial storage system (ProLab Diagnostics, Bromborough, Wirral, UK) at −70°C. *C. difficile* strains 630 and M 120 were kindly provided as a gift by Professor Brendan Wren from the Department of Pathogen Molecular Biology at the London School of Hygiene and Tropical Medicine, London, UK. Strain ATCC 9689 was purchased from bioMérieux (Marcy l’Etoile, France).

The tested strains were thawed, plated on Columbia agar, and incubated at 37 °C for 48 h under anaerobic conditions. All isolates were cultivated in brain–heart infusion medium (BHI; Difco, Detroit, MI, USA) at 37 °C for 48 h under anaerobic conditions.

### 4.2. Bee Products

MGO™ Manuka honey 550+, certified to contain at least 550 mg/kg of methylglyoxal, was obtained from Manuka Health, New Zealand. Goldenrod honey (*Solidago virgaurea*), pine herb honey (*Pinus sylvestris* L.), and bee bread (a combination of flower pollen, honey, royal jelly, and bee enzymes) were obtained from an apiary in southern Poland.

All bee products were stored at an appropriate temperature and were used for research immediately after harvesting. Stock solutions were prepared by dissolving in BHI medium and then filtered using 0.2-µm syringe filters (Corning, Corning, NY, USA).

### 4.3. Determination of the Minimum Inhibitory and Minimum Bactericidal Concentrations of Bee Products for C. difficile Strains Belonging to Different PCR-RTs

The minimum inhibitory concentration (MIC) for the growth of bacteria and the minimum bactericidal concentration (MBC) for the tested *C. difficile* strains were assessed according to the methodology described previously [62]. A 96-well titration plate (Nunc, Roskilde, Denmark) was used for the study. The tested products were dissolved in the BHI medium at the concentrations of 100%, 50%, 25%, 12.5%, 6.25%, 3.125%, and 1.56%, while bee bread was at the concentrations of 200 mg/L, 100 mg/L, 50 mg/L, 25 mg/L, and 12.5 mg/L. Next, 20 mL of a 3.0 McFarland suspension of tested strain with 0.85% NaCl was added to the wells of the titration plate containing 180 µL of the tested concentrations of bee products and incubated at 37 °C for 48 h under anaerobic conditions. The positive control was the BHI medium with 20 µL of suspension of *C. difficile* adjusted to 3.0 McFarland, and the negative control was the BHI medium. All strains were tested in triplicate. Following incubation, optical density at 600 nm was measured using a microplate reader (Bio-Rad, Hercules, CA, USA). The MBC was determined by plating the cell suspensions in 96-well plates used for MIC tests onto Columbia agar containing 5% sheep blood (Becton Dickinson, Heidelberg, Germany) and incubated at 37 °C for 48 h under anaerobic conditions. The bacterial growth was then visually observed.

### 4.4. Cell Cultures

Three human epithelial cell lines were used in the study: HT-29 passaged 15–25 times prior to use (from the cell-line library at the Anaerobic Laboratory, Department of Medical Microbiology); HT-29 MTX passaged 5–15 times (continuous lines of human colorectal adenocarcinoma cells: non-mucosa, and mucosa, respectively) (European Collection of Authenticated Cell Cultures, ECACC, Salisbury, UK); and CCD 841 CoN passaged 5–15 times (continuous line of human normal colon epithelial cells; American Type Culture Collection, ATCC, Manassas, VA, USA). The cell lines were stored in the cell bank at −196 °C. Cells were cultured in Dulbecco’s Modified Eagle Medium (DMEM; Lonza, Walkersville, MD, USA) with 4.5 g/L glucose and L-glutamine with addition of 10% heat-inactivated (56 °C, 30 min) fetal bovine serum (FBS) (Sigma-Aldrich, Saint Louis, MI, USA) and antibiotics: streptomycin, 100 μg/mL, penicillin, 100 U/mL, and amphotericin B, 250 μg/mL (Sigma-Aldrich, Saint Louis, MI, USA). Cells were thawed, replenished with culture fluid, and centrifuged at 1500× *g* for 5 min. Then, the growth medium was added to the cell pellet and transferred to 75-cm^2^ culture bottles (Corning, Corning, NY, USA) and supplemented with DMEM culture fluid (Lonza, Walkersville, MD, USA) and relative air humidity of 95%. The supernatant medium was changed every three days.

To ensure the continuity of the cell culture and to obtain the appropriate number of cells for the study, the cell-line cultures were passaged when reaching a confluence of 90%. The medium above the growth surface was harvested and washed with phosphate buffer saline (PBS; Lonza, Basel, Switzerland), then the cells were trypsinized with EDTA (0.25%) (Sigma-Aldrich, Saint Louis, MI, USA) at 37 °C for 10 min. The trypsin effect was inactivated by adding 10 mL of fresh medium with 10% FBS. Then, the contents of the bottle were centrifuged at 1500× *g* for 5 min, after which the supernatant was removed, and 1 mL of medium was added to the remaining pellet. The cells were then counted with a Thoma chamber and diluted to the desired amount in a new flask with a fresh medium and 10% FBS. The excess cells were transferred to a new culture vessel. Cells intended for the experiment of bacterial adhesion to colon cells were cultured by means of the above method using 24-well plates in an incubator with a flow of 5% CO_2_ and a temperature of 37 °C. On the day before the experiment, the culture fluid was replaced with a new one, without the addition of antibiotics. Experiments were performed on mature cells, which were 15-days-after-seeding HT-29 and CCD 841 CoN cells and 21-days-after-seeding HT-29 MXT cells, all passaged 15–25 times.

### 4.5. The Effect of Bee Products on the Adhesion of the Tested C. difficile Strains to Cell Lines

The 24-well plates of the 15-day culture of HT-29, HT-29 MTX, and CCD 841 CoN cell lines were washed twice with PBS buffer. After rinsing, the wells were supplemented with 400 μL of DMEM culture fluid with 1% content of individual tested bee products and incubated for 4 h at 37 °C with a flow of 5% CO_2_. Then, 100 μL of the suspension of the tested bacteria was added to the wells and incubated for another hour. The inoculum consisted of bacterial suspensions with a density of 3.0 on the McFarland scale prepared from 24 h bacterial cultures on a solid medium. The blank was the inoculum cell culture without the addition of bee products. After 1 h, the medium was harvested and washed twice with PBS buffer, then the cells were trypsinized for 10 min at 37 °C. The trypsin effect was stopped by adding 500 µL of medium with 10% FBS. The content of each well in the amount of 100 μL was transferred to sterile Eppendorf tubes containing 900 μL of sterile water, and 20 μL was spread on a solid medium with 5% sheep blood and incubated for 48 h at 37 °C under anaerobic conditions. The colonies were counted and averaged [27].

### 4.6. Testing the Formation of Biofilm of C. difficile Strains under the Influence of Bee Products

In the strains grown on Columbia agar, one colony was excised and transferred to BHI liquid medium. All *C. difficile* strains were incubated overnight in a BHI medium at 37 °C under anaerobic conditions. The study was performed on a 96-well titer plate. The suspension of bee products in BHI at a volume of 180 µL was added to the wells at concentrations of 100%, 50%, 25%, 12.5%, 6.25%, 3.125%, 1.56%, and 20 µL of overnight cultures of the test strains. Wells with BHI broth without the inoculum were used as controls. The plates were incubated at 37 °C for 48 h under anaerobic conditions for biofilm formation. After 48 h, the liquid phase was aspired using a sterile pipette, washed twice with PBS, and air dried at 37 °C for 15 min. Each well was then stained with crystal violet (Analab, Warsaw, Poland) for 10 min. The crystal violet was removed, and the wells were washed eight times with PBS. After air drying for 15 min at 37 °C, the crystal violet within the biofilms was dissolved in ethanol, and the absorbance was measured at 620 nm (A_620_) using a Bio-Rad 550 Microplate Reader (Bio-Rad, Hercules, CA, USA). All strains were tested six times. The average values for each *C. difficile* strain were calculated.

### 4.7. Confocal Laser Scanning Microscopy

The effects of bee products at different concentrations were visualized in culture with five different *C. difficile* strains, namely, 630 and M 120, and three randomly selected strains with representative RTs: 1468/12 (RT027), 25694/12 (RT023), and 2628/12 (RT176). Biofilms were grown on sterile 10 mm diameter glass-bottom dishes (Nunc, Roskilde, Denmark). Biofilm culture was prepared by dilution of overnight cultures of *C. difficile* in fresh BHI and BHI medium with subinhibitory doses of bee products: for strain 630, BHI medium supplemented with 25% of goldenrod honey, 3.125% of Manuka honey 550+, 12.5% of pine honey, and 50 mg/L of bee bread; for strain M 120, BHI medium supplemented with 6.25% of goldenrod honey, 0.78% of Manuka honey 550+, 12.5% of pine honey, and 50 mg/L of bee bread; for strain 1468/12, BHI medium supplemented with 6.25% of goldenrod honey, 3.125% of Manuka honey 550+, 1.56% of pine honey, and 25 mg/L of bee bread; for strain 25694/12, BHI medium supplemented with 25% of goldenrod honey, 3.125% of Manuka honey 550+, 12.5% of pine honey, and 12.5 mg/L of bee bread; and for strain 2628/12, BHI medium supplemented with 1.56% of goldenrod honey and Manuka honey 550+, 0.78% of pine honey, and 50 mg/L of bee bread. As a result, 2500 µL of BHI-supplemented medium and 500 µL of overnight cultures of *C. difficile* strains were obtained. The control was the biofilm of the tested strain without bee products. The biofilms were allowed to grow for 48 h at 37 °C under anaerobic conditions. The mature biofilms were then washed twice using 10 mM MgSO_4_ before staining with acridine orange (10 µg/mL) for 30 min in the dark. The dishes were washed twice with 10 mM MgSO_4_. Imaging was performed using a Nikon A1R MP microscope with a Nikon Ti Eclipse series (Nikon, Tokyo, Japan), using a ×60 objective lens and immersion oil. Images were acquired at 2040 × 2048 pixels using a Z-step of 0.1 μm. Acridine orange was detected using an excitation wavelength of 488 nm and an emission wavelength of 500–550 nm. Images were processed and analyzed using NIS-Elements AR v. 4.10 software.

### 4.8. Statistical Analysis

Statistical analysis was performed using Statistica software (version 13, StatSoft, TIBCO Software, Warsaw, Poland). The effect of bee products on *C. difficile* adhesion and biofilm formation was assessed using the Kruskal–Wallis one-way analysis of variance (ANOVA) followed by the Dunn’s test for comparison. Differences in biofilm formation were assessed using ANOVA followed by the Tukey’s post-hoc test. Differences were considered as statistically significant for *p* values < 0.05.

## 5. Conclusions

The RT023 and RT027 strains deserve attention. The RT023 strains showed a high MIC sensitivity to all bee products and strongly adhered to the healthy CCD 841 CoN line and the HT-29 MTX tumor line. Similarly, the strains with RT027 were also sensitive to all bee preparations and adhered strongly to the HT-29 line. In both cases, strains with these RTs produced a thin biofilm.

Our research proved that bee products had differential activity against the tested strains of *C. difficile* belonging to different PCR-RTs.

## Figures and Tables

**Figure 1 molecules-27-07385-f001:**
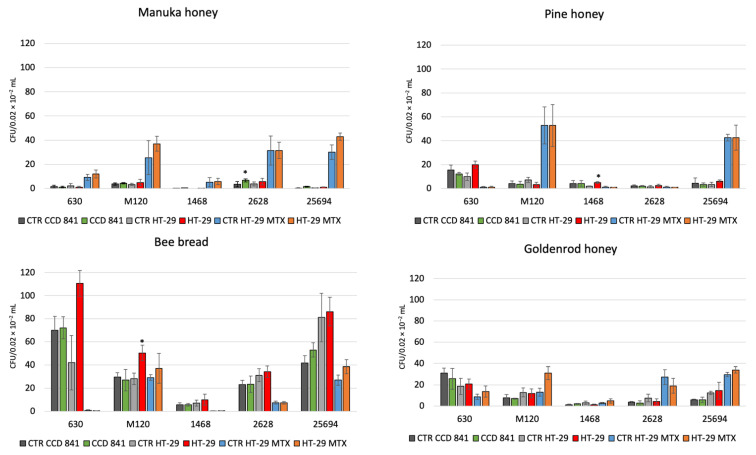
The effect of combination with 1%: Manuka honey 550+, goldenrod honey, pine honey, bee bread on the adhesion of *C. difficile* 630, M 120, 1468, 2628, 25694 to three human cell lines. Data are shown as means ± standard error. CFU—colony forming unit; *—*p* < 0.05.

**Figure 2 molecules-27-07385-f002:**
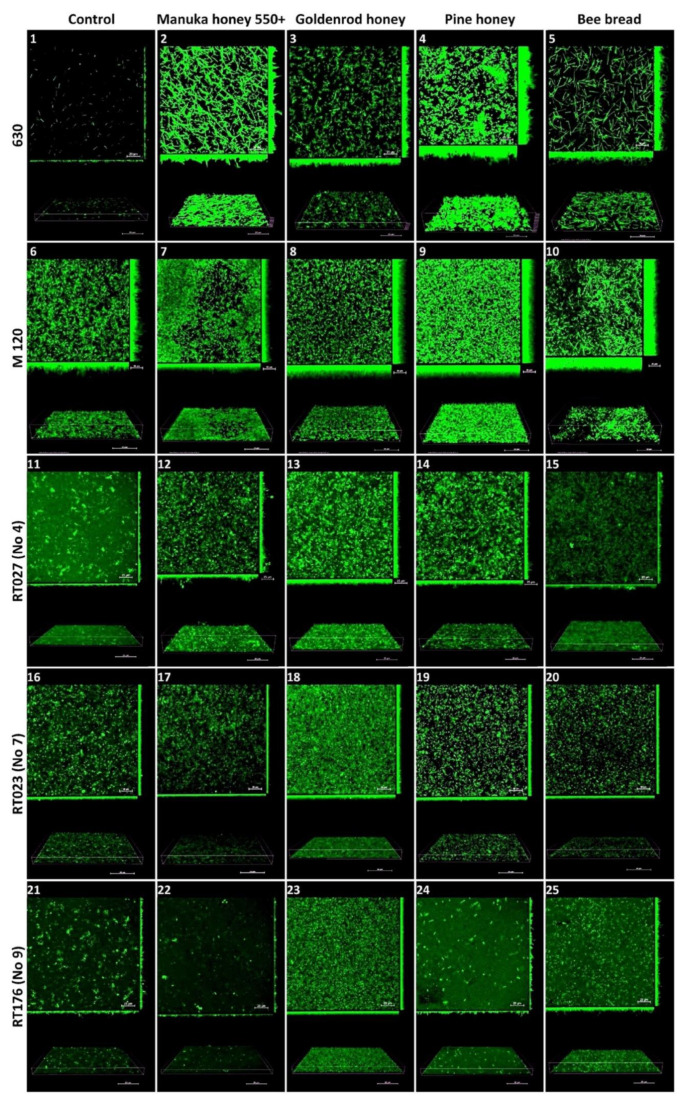
The effect of Manuka honey 550+, goldenrod honey, pine honey, and bee bread on biofilm formation by reference *C. difficile* strains 630 and M 120, and clinical strains RT027 (No 4), RT023 (No 7), and RT176 (No 9). Representative confocal microscopy images of horizontal (xy) and vertical (xz) projections of *C. difficile* biofilm structures. Slices view with maximum intensity projection. Legend: Control-630 (**1**), M 120 (**6**), RT027 (**11**), RT023 (**16**), RT176 (**21**); Manuka honey 550+ 630 (**2**), M 120 (**7**), RT027 (**12**), RT023 (**17**), RT176 (**22**); Goldenrod honey-630 (**3**), M 120 (**8**), RT027 (**13**), RT023 (**18**), RT176 (**23**); Pine honey-630 (**4**), M 120 (**9**), RT027 (**14**), RT023 (**19**), RT176 (**24**); Bee bread-630 (**5**), M 120 (**10**), RT027 (**15**), RT023 (**20**), RT176 (**25**).

**Table 1 molecules-27-07385-t001:** Minimum inhibitory concentrations (MIC) and minimum bactericidal concentrations (MBC) of Manuka honey 550+, goldenrod honey, pine honey, and bee bread on *C. difficile* strains.

Strain No.	RT	MIC% of Manuka Honey 550+	MBC% of Manuka Honey 550+	MIC% of Goldenrod Honey	MBC% of Goldenrod Honey	MIC% of Pine Honey	MBC% of Pine Honey	MIC (mg/L) of Bee Bread	MBC (mg/L) of Bee Bread
1 630	012	6.25	6.25	50	≥100	25	≥100	100	≥200
2 ATCC 9689	001	6.25	6.25	50	≥100	25	≥100	50	≥200
3 M 120	078	1.56	3.125	12.5	≥100	25	≥100	100	≥200
4 1468/12	027	6.25	6.25	12.5	50	3.125	12.5	50	≥200
5 2292	027	6.25	6.25	25	≥100	6.25	≥100	50	≥200
6 2333/06	023	6.25	6.25	25	≥100	12.5	≥100	25	≥200
7 25694/12	023	6.25	6.25	50	≥100	25	≥100	25	≥200
8 5291/12	176	1.56	3.125	1.56	50	1.56	50	50	≥200
9 2628/12	176	3.125	6.25	3.125	25	1.56	25	100	≥200
10 536/12	176	3.125	6.25	6.25	25	25	50	25	≥200
11 1974/12	176	3.125	6.25	12.5	25	25	50	25	≥200
12 3136/12	176	6.25	12.5	12.5	25	25	50	25	≥200

Legend: Strain No. 1, reference *C. difficile* 630; No. 2, reference *C. difficile* ATCC 9689; No. 3, control *C. difficile* M 120; No. 4–12, clinical *C. difficile* strains; RT, polymerase chain reaction ribotype.

**Table 2 molecules-27-07385-t002:** Average biofilm biomass ± standard deviation (crystal violet absorbance A620) by examined *C. difficile* strains with different concentrations of Manuka honey 550+.

Strain	CTR	MAN 100%	MAN 50%	MAN 25%	MAN 12.5%	MAN 6.25%	MAN 3.125%	MAN 1.56%
No 1 (630)	0.43 ± 0.02	0.10 ± 0.01 *	0.10 ± 0.02 *	0.11 ± 0.01 *	0.12 ± 0.01 *	0.17 ± 0.01 *	0.50 ± 0.01 *	0.51 ± 0.01 *
No 2 (ATCC 9689)	0.31 ± 0.02	0.10 ± 0.04 *	0.10 ± 0.02 *	0.15 ± 0.02 *	0.16 ± 0 *	0.16 ± 0.01 *	0.41 ± 0.06 *	0.62 ± 0.05 *
No 3 (M 120)	0.40 ± 0.02	0.14 ± 0.04 *	0.21 ± 0.02 *	0.14 ± 0.02 *	0.21 ± 0.02 *	0.23 ± 0.01 *	0.70 ± 0.01 *	0.55 ± 0.01 *
No 4 (1468/12) RT027	0.24 ± 0.02	0.20 ± 0.02	0.13 ± 0.02	0.16 ± 0.04	0.17 ± 0.07	0.21 ± 0.03	0.29 ± 0.07	0.29 ± 0.01
No 5 (2292) RT027	0.23 ± 0.03	0.10 ± 0 *	0.11 ± 0.02 *	0.11 ± 0.02 *	0.14 ± 0.02 *	0.12 ± 0.02 *	0.23 ± 0.07	0.33 ± 0.01 *
No 6 (2333/06) RT023	0.22 ± 0.02	0.09 ± 0.01 *	0.10 ± 0.03	0.11 ± 0.01 *	0.11 ± 0.02	0.15 ± 0.02 *	0.16 ± 0.04	0.25 ± 0.07
No 7 (25694/12) RT023	0.24 ± 0.03	0.09 ± 0.02 *	0.10 ± 0 *	0.11 ± 0.02 *	0.15 ± 0.02 *	0.18 ± 0.02 *	0.34 ± 0.03	0.33 ± 0.02
No 8 (5291/12) RT176	0.18 ± 0.04	0.09 ± 0.01 *	0.11 ± 0 *	0.17 ± 0.03 *	0.14 ± 0.02	0.16 ± 0.03	0.19 ± 0.3	0.20 ± 0.05
No 9 (2628/12) RT176	0.23 ± 0.06	0.09 ± 0.02	0.13 ± 0.03	0.11 ± 0.01	0.17 ± 0.03	0.13 ± 0.02	0.20 ± 0	0.29 ± 0.05
No 10 (536/12) RT176	0.21 ± 0.01	0.10 ± 0.01 *	0.12 ± 0.01 *	0.10 ± 0.02 *	0.11 ± 0.02 *	0.38 ± 0.02 *	0.37 ± 0.03 *	0.38 ± 0.03 *
No 11 (1974/12) RT176	0.12 ± 0.01	0.10 ± 0.01	0.12 ± 0.01	0.11 ± 0.01	0.10 ± 0.01	0.11 ± 0.01	0.10 ± 0.01	0.11 ± 0.01
No 12 (3136/12) RT176	0.14 ± 0.02	0.10 ± 0.01 *	0.12 ± 0.01	0.12 ± 0.01	0.17 ± 0.01	0.12 ± 0.01	0.12 ± 0.01	0.10 ± 0.01

CTR—control; MAN—Manuka honey 550+; *—statistically significant.

**Table 3 molecules-27-07385-t003:** Average biofilm biomass ± standard deviation (crystal violet absorbance A620) by examined *C. difficile* strains with different concentrations of goldenrod honey.

Strain	CTR	GOL 100%	GOL 50%	GOL 25%	GOL 12.5%	GOL 6.25%	GOL 3.125%	GOL 1.56%
No 1 (630)	0.43 ± 0.04	0.11 ± 0.01 *	0.21 ± 0.06 *	0.33 ± 0.1	0.39 ± 0.14	0.43 ± 0.07	0.43 ± 0.03	0.45 ± 0.03
No 2 (ATCC 9689)	0.28 ± 0.02	0.09 ± 0.02 *	0.10 ± 0.02 *	0.17 ± 0.02	0.27 ± 0.06	0.39 ± 0.06	0.41 ± 0.06	0.53 ± 0.1 *
No 3 (M 120)	0.43 ± 0.02	0.09 ± 0.02 *	0.16 ± 0.02 *	0.26 ± 0.04 *	0.34 ± 0.05	0.54 ± 0.08	0.49 ± 0.04	0.50 ± 0.07
No 4 (1468/12) RT027	0.33 ± 0.12	0.08 ± 0.02 *	0.14 ± 0.02 *	0.15 ± 0.02 *	0.21 ± 0.02	0.21 ± 0.02	0.31 ± 0.02	0.25 ± 0.09
No 5 (2292) RT027	0.20 ± 0.02	0.08 ± 0.02 *	0.09 ± 0.02 *	0.14 ± 0.03	0.21 ± 0.02	0.27 ± 0.04	0.33 ± 0.02 *	0.27 ± 0.02 *
No 6 (2333/06) RT023	0.23 ± 0.01	0.09 ± 0.03	0.08 ± 0	0.24 ± 0.04	0.23 ± 0.06	0.38 ± 0.05	0.40 ± 0.12	0.30 ± 0.03
No 7 (25694/12) RT023	0.16 ± 0.05	0.08 ± 0	0.07 ± 0	0.17 ± 0	0.27 ± 0.06	0.34 ± 0.13 *	0.23 ± 0.15	0.22 ± 0.01
No 8 (5291/12) RT176	0.16 ± 0.03	0.06 ± 0.01 *	0.10 ± 0.02 *	0.10 ± 0.05 *	0.09 ± 0.01 *	0.18 ± 0.01	0.18 ± 0.01	0.18 ± 0.01
No 9 (2628/12) RT176	0.18 ± 0.04	0.10 ± 0 *	0.10 ± 0 *	0.12 ± 0.01	0.12 ± 0.04 *	0.17 ± 0.01	0.17 ± 0.01	0.18 ± 0.01
No 10 (536/12) RT176	0.21 ± 0.01	0.05 ± 0.01 *	0.18 ± 0.02	0.33 ± 0.12	0.41 ± 0.02 *	0.40 ± 0.02 *	0.41 ± 0.04 *	0.40 ± 0.05 *
No 11 (1974/12) RT176	0.12 ± 0.01	0.10 ± 0.01	0.12 ± 0.01	0.11 ± 0.01	0.10 ± 0.01	0.11 ± 0.01	0.10 ± 0.01	0.11 ± 0.01
No 12 (3136/12) RT176	0.15 ± 0.02	0.10 ± 0.01 *	0.10 ± 0.01 *	0.12 ± 0.01 *	0.12 ± 0.02	0.12 ± 0.01 *	0.11 ± 0.01 *	0.13 ± 0.01

CTR—control; GOL—goldenrod honey; *—statistically significant.

**Table 4 molecules-27-07385-t004:** Average biofilm biomass ± standard deviation (crystal violet absorbance A620) by examined *C. difficile* strains with different concentrations of pine honey.

Strain	CTR	PIN 100%	PIN 50%	PIN 25%	PIN 12.5%	PIN 6.25%	PIN 3.125%	PIN 1.56%
No 1 (630)	0.24 ± 0.02	0.11 ± 0.01 *	0.12 ± 0.06 *	0.16 ± 0.02 *	0.18 ± 0.03	0.22 ± 0.02	0.21 ± 0.01	0.39 ± 0.03 *
No 2 (ATCC 9689)	0.38 ± 0.02	0.09 ± 0.01 *	0.10 ± 0.02 *	0.17 ± 0.02 *	0.17 ± 0.02 *	0.18 ± 0	0.24 ± 0.16	0.28 ± 0.18
No 3 (M 120)	0.38 ± 0.01	0.09 ± 0.02 *	0.11 ± 0.02 *	0.14 ± 0.01 *	0.19 ± 0.01 *	0.30 ± 0.12	0.33 ± 0.14	0.42 ± 0.07
No 4 (1468/12) RT027	0.11 ± 0.01	0.07 ± 0.01 *	0.07 ± 0.01 *	0.08 ± 0.01 *	0.08 ± 0.01 *	0.08 ± 0.01 *	0.08 ± 0.01 *	0.08 ± 0.01 *
No 5 (2292) RT027	0.12 ± 0.02	0.08 ± 0.01	0.07 ± 0.01	0.14 ± 0.01	0.15 ± 0.01	0.17 ± 0.01	0.25 ± 0.12 *	0.23 ± 0.02 *
No 6 (2333/06) RT023	0.1 ± 0.01	0.07 ± 0.01 *	0.09 ± 0	0.09 ± 0.01	0.1 ± 0.01	0.09 ± 0.01	0.08 ± 0.01	0.09 ± 0.01
No 7 (25694/12) RT023	0.11 ± 0.01	0.08 ± 0.01	0.07 ± 0.01	0.09 ± 0.02	0.17 ± 0.03	0.30 ± 0.16 *	0.2 ± 0.13	0.21 ± 0.13
No 8 (5291/12) RT176	0.11 ± 0.01	0.11 ± 0.02	0.09 ± 0.01	0.12 ± 0.02	0.13 ± 0.01	0.12 ± 0.01	0.12 ± 0.01	0.11 ± 0.01
No 9 (2628/12) RT176	0.12 ± 0.01	0.10 ± 0.01	0.10 ± 0	0.12 ± 0	0.12 ± 0.01	0.13 ± 0.02	0,18 ± 0.01 *	0,17 ± 0.02 *
No 10 (536/12) RT176	0.18 ± 0.02	0.10 ± 0.01	0.18 ± 0.02	0.38 ± 0.12 *	0.41 ± 0.12 *	0.36 ± 0.07 *	0.39 ± 0.06 *	0.35 ± 0.05
No 11 (1974/12) RT176	0.11 ± 0.01	0.1 ± 0.01	0.1 ± 0.01	0.1 ± 0.01	0.13 ± 0.01 *	0.1 ± 0.01	0.1 ± 0.01	0.1 ± 0.01
No 12 (3136/12) RT176	0.11 ± 0.02	0.1 ± 0.02	0.1 ± 0.01	0.07 ± 0.01	0.1 ± 0.01	0.1 ± 0.01	0.1 ± 0.02	0.1 ± 0.02

CTR–control; PIN—pine honey; *—statistically significant.

**Table 5 molecules-27-07385-t005:** Average biofilm biomass ± standard deviation (crystal violet absorbance A620) by examined *C. difficile* strains with different concentrations of bee bread.

Strain	CTR	BEB 200%	BEB 100%	BEB 50%	BEB 25%	BEB 12.5%	BEB 6.25%	BEB 3.125%	BEB 1.56%
No 1 (630)	0.11 ± 0	0.17 ± 0.01 *	0.18 ± 0.02 *	0.11 ± 0.01	0.15 ± 0.01	0.10 ± 0.01	0.12 ± 0.01	0.12 ± 0.01	0.11 ± 0.01
No 2 (ATCC 9689)	0.11 ± 0.01	0.18 ± 0.01 *	0.19 ± 0.01 *	0.17 ± 0.02 *	0.11 ± 0.02	0.11 ± 0	0.12 ± 0.01	0.12 ± 0.01	0.12 ± 0.01
No 3 (M 120)	0.31 ± 0.11	0.15 ± 0.01 *	0.19 ± 0.01	0.12 ± 0.01 *	0.45 ± 0.07	0.31 ± 0.01	0.33 ± 0.03	0.32 ± 0.03	0.32 ± 0.03
No 4 (1468/12) RT027	0.11 ± 0.01	0.13 ± 0.01	0.16 ± 0.01	0.18 ± 0.01	0.15 ± 0.01	0.15 ± 0.01	0.17 ± 0.01	0.18 ± 0.01	0.18 ± 0.01
No 5 (2292) RT027	0.15 ± 0.02	0.13 ± 0.01	0.08 ± 0.01	0.07 ± 0.01	0.14 ± 0.01	0.15 ± 0.01	0.17 ± 0.01	0.25 ± 0.12	0.23 ± 0.02
No 6 (2333/06) RT023	0.25 ± 0	0.1 ± 0.01 *	0.17 ± 0.1	0.21 ± 0.12	0.2 ± 0.1	0.17 ± 0.07	0.3 ± 0.04	0.32 ± 0.02	0.21 ± 0.09
No 7 (25694/12) RT023	0.18 ± 0.01	0.16 ± 0.01	0.18 ± 0.01	0.18 ± 0.01	0.17 ± 0.05	0.16 ± 0.03	0.18 ± 0.01	0.17 ± 0.01	0.18 ± 0.04
No 8 (5291/12) RT176	0.11 ± 0.01	0.15 ± 0.01	0.16 ± 0.01 *	0.14 ± 0.03	0.13 ± 0.01	0.13 ± 0.01	0.12 ± 0.01	0.12 ± 0.01	0.11 ± 0.01
No 9 (2628/12) RT176	0.14 ± 0.01	0.13 ± 0.01	0.18 ± 0.03 *	0.10 ± 0.01	0.12 ± 0.01	0.12 ± 0.01	0.13 ± 0.02	0.13 ± 0.01	0.13 ± 0.01
No 10 (536/12) RT176	0.14 ± 0.01	0.13 ± 0.01	0.14 ± 0.03	0.14 ± 0.01	0.14 ± 0.01	0.14 ± 0.01	0.14 ± 0.02	0.14 ± 0.01	0.13 ± 0.01
No 11 (1974/12) RT176	0.14 ± 0.01	0.15 ± 0.01	0.14 ± 0.03	0.14 ± 0.01	0.14 ± 0.03	0.09 ± 0.01	0.14 ± 0.02	0.14 ± 0.01	0.13 ± 0.01
No 12 (3136/12) RT176	0.14 ± 0.03	0.14 ± 0.01	0.14 ± 0.03	0.12 ± 0 *	0.14 ± 0.03	0.14 ± 0.01	0.14 ± 0.02	0.14 ± 0.01	0.13 ± 0.01

CTR—control; BEB—bee bread; *—statistically significant.

## Data Availability

The data underlying this article will be shared on reasonable request to the corresponding author.

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
