# Peer review of "The Effect of Selected Bee Products on Adhesion and Biofilm of Clostridioides difficile Strains Belonging to Different Ribotypes"

_molecules, 2022, doi:10.3390/molecules27217385_

Round 1

Reviewer 1 Report

Manuscript ID : molecules-1963897

Title: Antiadhesive and antibiofilm activities of selected bee products against Clostridioides difficile strains belonging to different ribotypes

Authors: Dorota Wultańska * , Bohdan Paterczyk , Julita Nowakowska , Hanna Pituch *

In this manuscript authors reported the description of the antibacterial and antibiotic activity of bee products against C. difficile strains. The minimum inhibitory concentration (MICs) of Manuka honey 550+, goldenrod honey, pine honey, and bee bread were determined by the broth dilution method. C. difficile adhesion to HT29, HT29-MTX, and CCD 841 CoN cell lines was assessed. Biofilm was cultured in titration plates and visualized by confocal microscopy. The MICs of Manuka honey for C. difficile 630 and ATCC 9689 strains and control strain, M120, were 6.25%, 6.25%, and 1.56% (v/v), respectively; of goldenrod honey, 50%, 50%, and 12.5%, respectively; of pine honey, 25%, 25%, and 25%, respectively, bee bread, 100 mg/L, 50 mg/L, and 100 mg/L, respectively. Manuka honey (1%) increased adhesion of C. difficile RT176 strains, and one strain of RT023 to the CCD841 cell line. Pine honey (1%) increased RT027 adhesion to the HT29 cell line. Manuka honey, pine honey, and bee bread at subinhibitory concentrations increased the adhesion of C. difficile. Authors concluded that bee products are active against the tested strains of C. difficile.

However, there are some modifications required to be done before it is accepted for publication. The following are the specific comments on the manuscript:

Major concern:

The major concern of this manuscript is related to the chemical composition of the tested bee products. Antibacterial and antiadhesion activities of the tested bee products should be correlated with the phytochemical composition obtained.

All data from figures 2, 3, and 4 should be better summarized in tables.

I think that this paper can not be accepted in its current form, authors should add results related to the chemical composition of the selected bee product and a correlation between the biological activities tested (antibacterial/antibiofilm) and the phytoconstituents identified have to be performed.

Reviewer 2 Report

The authors intended to characterize the influence of selected bee products on the growth, adhesion, and biofilm formation of Clostridioides difficile strains with different ribotypes. In general, the manuscript is clearly structured and the methods used to obtain the results are well-defined. However, several aspects need to be clarified and revisions would be appropriate.

Please consider the following points for the revision of your manuscript:

The whole study is called “Antiadhesive and antibiofilm activities of selected bee products against Clostridioides difficile strains belonging to different ribotypes ”. Although that name refers to the antiadhesive properties of selected bee products, the results in the study showed only that manuka honey, pine honey, and bee bread (at subinhibitory concentrations) increased the adhesion of C. difficile. Unfortunately, these results are not even mentioned in the discussion section for a more detailed comment.

In chapter 2.2 Effect of bee products on adhesive properties of C. difficile it is stated that C. difficile strains showed adhesion to the used cell lines. But the results showed quite big differences in the rate of adhesion even in the control samples between different experiments e.g. results of CTRL CCD 841 samples in charts “Manuka honey” and “Bee bread”. Would the authors please provide a comment on that?

Inconsistent designations appear in the text: CCD 841 and CCD841; HT-29 and HT29; RT176 and RT 176

The charts in Figure 1 are not the same size or formatting, and the description of the y-axis is missing. The legend for this figure is abnormally brief and so insufficient. Among other things, there is no complete explanation of the nature of the samples shown in the graphs. It would be useful to add this information. The legends for Figures 2–5 are similarly brief and not descriptive enough.

The sentence on lines 35–37 is not clearly formulated. It can be understood that mentioned antibiotics, after eradicating the natural microflora, also suppress the growth of C. difficile while these antibiotics as cephalosporins, clindamycin, and fluoroquinolones inhibit the growth of healthy microflora and thus increase the risk of C. difficile infection.

On line 251 is stated that “Honey, which also stimulates inflammatory cytokines…” This statement is based on a publication describing honey's wound-healing properties. However, in general, anti-inflammatory activity is confirmed for honey where mainly honey flavonoids can suppress the production of pro-inflammatory cytokines and also inhibit the activity of enzymes that regulate the process of inflammation. Therefore such a simplistic statement as presented in the article may be misleading to the readers.

Round 2

Reviewer 1 Report

Dear Authors

Please revise figure 1 : you can modify the scale of the axis to make figure 1 more clear

All the best

Author Response

Dear reviewer,

thank you for this suggestion. As recommended by the previous reviewer, we have adopted a homogeneous scale in all the charts in Figure 1. In bee bread the values reached about 120 therefore in each chart the maximum was set at 120 to not confuse the readers. If you think that the scale should be changed then we will make the necessary changes.